# Understanding Atherosclerotic Plaque Cellular Composition: Recent Advances Driven by Single Cell Omics

**DOI:** 10.3390/cells14110770

**Published:** 2025-05-23

**Authors:** Esra Cetin, Anne-Catherine Raby

**Affiliations:** Wales Kidney Research Unit, Division of Infection and Immunity, School of Medicine, https://ror.org/03kk7td41Cardiff University, Cardiff CF14 4XN, UK

**Keywords:** atherosclerosis, single-cell omics

## Abstract

Atherosclerosis, a chronic inflammatory condition of the arterial cell wall, is the leading cause of cardiovascular disease (CVD) and death. It is characterised by the accumulation of lipid and immune cell-rich plaques in the arterial intima, which is driven by a dysregulated immune response to cholesterol-containing lipoproteins at the lesion site. Initially thought to be driven by passive lipid accumulation, atherosclerosis is now recognised as a complex process encompassing a multitude of inflammatory and remodelling mechanisms, driven by both immune cells (e.g., macrophages, T-cells, B-cells, antigen-presenting cells) and non-immune cells (smooth muscle cells, endothelial cells). With the development of single-cell transcriptomic and proteomic technologies, a previously inconceivable wealth of data has been generated in an attempt to better understand the pathophysiology of the disease and identify novel avenues for the development of targeted therapeutic interventions. This review provides an overview of the latest findings in the field obtained using single-cell technologies, with a focus on the major cell types present at the atherosclerotic plaque site, their suggested repartition in subsets, as well as their predicted function(s) within the complex processes that interplay to drive atherosclerotic disease. We conclude by highlighting the discrepancies and areas of consensus brought about by these studies and briefly discuss the likely future advances that will come from the continuous development and improvement of single-cell omics technologies.

## Introduction

1

Atherosclerosis is a chronic inflammatory condition of the arterial cell wall, driven by an immune response to atherogenic lipids. It is characterised by the development of lipid- and immune cell-rich intimal plaques, starting early in life and progressing over time, which may grow to restrict blood flow or rupture, the latter leading to life-threatening thrombosis [1,2]. Thus, atherosclerosis is the main underlying cause of cardiovascular disease (CVD) and the leading cause of death worldwide. Atherogenesis involves the interplay of several pathways, including inflammation and tissue remodelling, as well as a number of immune and non-immune cells, such as macrophages, T-cells, B-cells, antigen-presenting cells (APCs), smooth muscle cells (SMCs), and endothelial cells (ECs). Cell trans-differentiation has also been documented as part of the atherosclerotic process, notably the transition of SMCs to fibroblast-like, macrophage-like, or osteogenic-like cells [3]. These complex interplays render the precise understanding of the pathophysiology of atherosclerosis challenging.

While cellular composition profiles of the atherosclerotic plaque have been proposed based on marker proteins [4], a detailed description of the populations and sub-populations of cells involved, their origin, and their specific roles in the pathophysiology of the disease remain to be established. Recent technological advances in transcriptomic and proteomic analyses, notably by allowing resolution at the single-cell level, have led to a boom of studies aiming to shed light on the complexity of atherosclerosis development and progression. Single-cell omic technologies offer unprecedented resolution and sensitivity by enabling the in-depth study of individual cells, thus revealing cellular heterogeneity and dynamic cellular states, allowing the identification of rare cell populations and the study of cell–cell communication.

Here, we provide an overview of the recent advances in the characterisation and understanding of the pathophysiology of atherosclerosis that were brought about by single-cell transcriptomic and proteomic analyses (summary in Table 1). We first discuss the different cell types identified within atherosclerotic lesions and follow with a detailed description of newly identified cellular subtypes and their proposed role in driving (or controlling) the disease. Apparent discrepancies and contradictions between studies are also discussed, and we conclude by considering remaining gaps in understanding and potential future directions.

## Overview of Atherosclerotic Plaque Cellular Composition

2

The difficulty in tracking the progression of atherosclerosis in human studies has led to a heavy reliance on the use of animal models to elicit an improved understanding of the process. While these models may not fully recapitulate human pathology, they have provided an essential tool for the study of the mechanisms driving atherosclerosis and the evaluation of therapeutic strategies against it. While discussing the limitations of in vivo models of atherosclerosis is beyond the scope of this review, these have been extensively described elsewhere [16,17].

Prior to the development of single-cell transcriptomics, one of the most in-depth characterisations of immune cell proportions in the atherosclerotic plaque was obtained by cytometry by time of flight (CyTOF) analysis of murine aortas, using a panel of 35 markers [18]. In *ApoE*^−/−^ mice, 13 leukocyte populations were identified. The most represented leukocyte (CD45^+^) subset was macrophages (~55%), followed by conventional dendritic cells (cDC, 2 subsets, ~7–9% in total), T-cells (~7%, 3 subsets CD4^+^, CD8^+^, γδ), B-cells (~5–7%), monocytes (~3–7%), neutrophils (~2–4%) and innate lymphoid cells (~1.5%). Small populations (<1%) of plasmacytoid dendritic cells (pDC), eosinophils and NK cells were also identified. While these 13 clusters were detected in animals with different degrees of atherosclerosis severity (chow diet vs. High Fat Diet, HFD), increased plaque development due to HFD led to an increase in monocytes (both inflammatory Ly6C^high^ and patrolling Ly6C^low^), pDC, and a CD11c^+^ macrophage subset. Simultaneously, CD206^+^ CD169^+^ macrophages were significantly decreased, as was a subset of cDC.

Single-cell RNA sequencing (scRNAseq) of aortas obtained from the same *ApoE*^−/−^ mouse model (females only) identified 11 leukocyte populations, including 5× T-cell populations, 2× B-cells, 1× macrophage, 2× monocytes and 1× NK cells [6]. Of note, relative proportions were different from those established by CyTOF, with the most abundant leukocytes being T-cells (49%), followed by B-cells (29%), myeloid cells (22%) and NK cells (2%). Confirming the CyTOF findings by Cole et al., Winkels et al. also found increased proportions of macrophages in response to HFD but reported fewer monocytes, notably the LyC6^high^ subpopulation. Notably, overall proportions of T and B cells appeared mostly unaffected by HFD exposure, but there were changes in subpopulations, with increased atherosclerotic burden being associated with more Th17 T-cells and less memory and Th2 T-cells (see dedicated T-cell section below). Importantly, the use of an alternative approach based on deconvolution of bulk mRNA data from whole aortas revealed a cellular profile more similar to that described by Cole et al. and dominated by macrophages (~40%). Using deconvolution of bulk RNAseq from laser-microdissected aortic tissues, Winkels et al. reported that atherosclerotic lesions were predominantly populated by macrophages, monocytes, and T-cells, while most of the B cells were located in Artery Tertiary Lymphoid Organs (ATLO)-rich adventitial tissue [6]. The authors also used whole aorta scRNAseq to compare the cellular profiles of atherosclerosis in the 2 main models of the disease, namely *ApoE*^−/−^ and *LdlR*^−/−^ mice on HFD. While T-cells were found to be the dominating leukocyte population in both models, aortas from *LdlR*^−/−^ mice on HFD had twice the proportions of myeloid cells (macrophages and monocytes), and about 6 times fewer B-cells. Notably, aortic leukocyte proportions in chow diet-fed mice were similar between both mouse strains, suggesting differences in the response to HFD.

In line with the findings by Winkels et al., Cochain et al. identified 13 cell types in the *LdlR*^−/−^ + HFD mouse model (males only), including three macrophage populations (28.9%), four T-cell populations (28.3%), dendritic cells (DCs)/Monocyte-derived (Mo) DCs (14.9%) and monocytes (12.3%) [5]. The remaining leukocyte population (15.7%) was made up of B-cells, granulocytes and NK cells. Interestingly, five of these clusters were found exclusively in atherosclerotic aortas (i.e., not in chow-diet animals), namely CD8^+^ T cells, MoDCs/DCs, monocytes and two populations of macrophages. Of these 5 atherosclerosis-restricted populations, the proportion of macrophages, but not that of the other 4 cell types, further increased with atherosclerosis progression (12 weeks HFD, 28.9%; 20 weeks HFD, 49.6%). Similarly to Winkels et al., Cochain et al. also reported that macrophages represented a smaller proportion of immune cells (9.6%) in *ApoE*^−/−^, compared to *Ldlr*^−/−^, atherosclerotic aortas. Instead, there was a shift towards increased lymphocytic presence, with T-cells being the most prominent cells (38.2%), followed by B-cells (24.3%). This suggests that differences in the specific mechanisms driving atherosclerosis progression between the two models (e.g., differences in lipoprotein types or degree of oxidation [19]) may induce different immune responses and relative contributions of major immune cell types.

Winkels et al. also investigated the leukocyte composition of human atherosclerotic plaque by CyTOF analysis (35 markers) of a carotid endarterectomy sample (stent placement). This strategy identified 19 clusters, but only a small percentage of cells from myeloid lineages [6], in opposition to previous findings from histological analyses of human plaques, which established a predominance of macrophages. To assess a potential false-negative result due to exclusion of macrophages as part of the CyTOF process (e.g., lack of specific human macrophages markers, macrophage death during tissue processing), the authors applied again a deconvolution strategy to bulk transcriptomic analyses of 126 carotid plaques, and found 51% macrophages, in line with prior histology findings, as well as 19% T-cells (five clusters), 14% monocytes (two clusters) 11% B-cells (two clusters) and 5% NK cells [6]. Notably, most leukocyte populations (not CD4^+^CD8^+^ T-cells and NK-cells) were higher in plaques compared to PBMCs from the same patients, indicating a non-random enrichment at the plaque site.

Depuydt et al. analysed atherosclerotic arteries from 18 patients undergoing carotid endarterectomy as a treatment for cerebrovascular disease, coronary artery disease or peripheral arterial occlusive disease. Using scRNAseq, they observed 14 clusters, including 11 leukocyte clusters (87% of the total) and 3 non-immune cell clusters (2xECs and 1xSMCs, 13% of the total) [10]. T-cells were the predominant population (52%, 4 clusters), in line with Winkels’ CyTOF findings, but in opposition to those made by deconvolution of bulk transcriptomic analysis. Other identified immune populations were myeloid cells (18.5%), including macrophages and DCs, B cells (3.6%) and mast cells (1.1%) [10]. Importantly, and as opposed to Winkels et al., Depuydt et al. reported that the predominance of T-cells, compared to myeloid cells, was confirmed histologically in the corresponding endarterectomy samples. The predominance of T-cells within the human atherosclerotic plaque was also reported by Fernandez et al., whose CyTOF analysis of immune cells from 46 carotid endarterectomy samples identified main leukocyte populations in proportions similar to Winkels (CyTOF, unconfirmed by deconvolution of bulk RNA seq) and Depuydt. Specifically, T-cells (CD4^+^ or CD8^+^) accounted for about 65% of all leukocytes, macrophages for 10%, monocytes and DCs for 3% each and NK-cells for 5% [9]. Among these, CD8^+^ T-cells were the most increased in plaque compared to blood from the same patient, suggesting a preferential enrichment, while proportions of CD4+ T-cells were similar or slightly lower in plaque than blood. As expected, macrophages were not found in blood, while monocytes and NK cells were more predominant in blood than plaque [9].

It is noteworthy that, despite the significant differences in circulating leukocyte proportions between mice and humans, with higher percentages of lymphocytes in mice, and higher proportions of neutrophils in humans, macrophages and T-cells are reported as the most predominant plaque leukocyte populations in both species, with neutrophils being mostly absent from plaques (Table 2). Thus, while mouse models of atherosclerosis may not fully recapitulate the human pathology, proportions of different plaque leukocytes, at least, appear comparable between the 2 species.

## Main Atherosclerotic Cell Populations: Transcriptomic, Phenotypic, and Functional Heterogeneity

3

### Macrophages

3.1

As described previously, macrophages are one of the dominant immune cell populations in the atherosclerotic plaque and play a decisive role at all stages of the disease. A key feature of macrophages is their plasticity, which allows a fine-tuned response to specific microenvironmental stimuli. As such, the atherosclerotic lesion site provides a complex microenvironment, including notably activated cells, modified lipoproteins, pro-inflammatory mediators and dying cells. Lesional macrophages perform a wide range of functions that control atherosclerosis progression, and it has long been described that plaques contain several macrophage subpopulations [4,21]. Consistently, as many as nine different subsets of atherosclerotic macrophages were proposed in humans and mice, based on the expression of a number of protein markers, prior to the development of single-cell omics [22]. Interestingly, scRNAseq analyses of atherosclerotic aortas led to fewer clusters being identified overall, suggesting heterogenous macrophage subgroups, rather than truly different subpopulations.

In a study focused on deciphering macrophage heterogeneity at the site of atherosclerotic lesions, Cochain et al. identified three clusters of macrophages in aortas from *LdlR*^−/−^ mice, which they named Inflammatory (47%), Resident-like (34.4%), and TREM2^hi^ (18.6%) [5]. All macrophage subsets were characterised by the expression of *Adgre1, Csf1r, Fcgr1*, and *Cd68*. In single-cell differential expression analyses of each subset against the other 2, Inflammatory macrophages were found to highly express genes previously described as pro-inflammatory and/or proatherogenic (e.g., *Cxcl2, Tlr2, Ccl3, Il1b, Il1a, Nlrp3, Cebpb, Egr1, Phlda1*). However, despite aligning with a “pro-inflammatory” phenotype, this subset also displayed the strongest expression of feedback inhibitors of macrophage activation (*Nfkbiz, Nfkbid, Ier3*), including some with established anti-atherogenic functions (*Zpf36, Nr4a1*). The resident-like subset expressed genes previously associated with aortic resident macrophages (*F13a1, Lyve1*), together with genes described as M2-like (*Folr2, Cbr2, Mrc1*). This cluster also expressed chemokines (e.g., *Ccl24, Ccl9*) and other proatherogenic genes (e.g., *Pf4, Txnip*). Of note, the resident-like macrophage cluster in atherosclerotic aortas displayed a higher expression of *Ccr2*, encoding for the receptor for the monocyte chemoattractant CCL2 (also known as MCP-1), than in control aortas. This suggests that, while there are genuine resident macrophages in atherosclerotic aortas, some of the infiltrating macrophages may adopt a similar profile once in the plaque. TREM2^hi^ macrophages, in addition to abundantly expressing *Trem2*, had a unique gene expression profile, including several cathepsins (*Ctsd, Ctsb, Ctsz*), *Cd9, Spp1* (encoding osteopontin) and *Hvcn1*.

Differences in functions between the three clusters were investigated by gene ontology (GO) term analyses. Inflammatory macrophages were associated with the highest number of different functions and notably strongly preferentially contributed to the *Regulation of leukocyte adhesion to vascular endothelial cells* and the *Regulation of monocyte chemotaxis* pathways. Resident-like macrophages were the apparent biggest contributor to *Antigen processing and presentation of exogenous peptide antigen* via *MHC class II*, and TREM2^hi^ macrophages were enriched for a particular set of functions not found in the other 2 subsets, namely cellular catabolic processes, lipid metabolic processes, and oxidative stress. However, there were overlaps in several biological processes, including *Myeloid leukocyte activation* being shared by all macrophage subsets, and *Inflammatory responses* by Inflammatory and Resident-like macrophages.

Critically, the authors attempted to match previously identified markers of macrophage populations commonly considered to populate atherosclerotic plaques in mice (M1, M2, and Mox) to their three identified subsets. They found that their Inflammatory subset preferentially expressed genes encoding for M1 (IL1β, TNFα, CXCL-10, CXCL-2, CCL-2) and Mox (NRF2) markers, and their Resident-like subset genes encoding for M2 markers (MRC-1, FOLR2, F13α). However, the expression of these markers was heterogeneous within a given cluster and overlapped between clusters, and no pattern could be found to assign TREM2^h^i macrophages to previously proposed subsets.

Importantly, although Cochain et al. reported macrophages to be less abundant in *ApoE*^−/−^, compared to *LdlR*^−/−^, animals on HFD, the genes characterizing the three subsets were still present, and subset proportions were similarly distributed in both mouse models [5].

In *ApoE*^−/−^ mice on HFD, Winkels et al. identified a single cluster of macrophages (*Csf1r, Adgre1, Cd68*) by scRNAseq of atherosclerotic aortas, while two clusters were identified in animals on normal chow [6]. These clusters expressed *Cxcr1*, indicating their resident and self-renewing state while having a differential expression of *Lyve1* (resident) and *Ly6c1* (recruited). This suggests, in line with the study by Cochain et al., that recruited macrophages may adopt a resident-like phenotype once in the plaque. Interestingly, the expression of *Cx3cr1* and *Lyve1* on aortic macrophages was maintained with increased atherosclerotic burden (i.e., in HFD-fed animals), raising the question of the extent to which the atherosclerotic macrophage pool is replenished from infiltrating monocytes, as previously described [4], or from in situ proliferation.

In gene enrichment analyses, macrophages were found associated with several curated pathways, such as *Lipid metabolism* (e.g., *Anxa5, Abcg1, Ldlr*), *Chemokines* (e.g., *Cxcl10, Cxcl12, Cxcl1, Ccl2, Ccl4*), *Apoptosis* (e.g., *Chp2, Prkar1b*), and *Inflammasome activation* (e.g., *Nlrp3, Casp1*). Ingenuity Pathway Analysis (IPA) of differentially expressed genes between chow-fed and HFD-fed animals revealed that HFD macrophages up-regulated iNOS-associated signaling, and pro-atherogenic Interferon regulatory transcription factors (IRF). In contrast, HFD decreased plaque-stabilising TGF-β signalling in macrophages. Like Cochain, Winkels also reported discrepancies between transcriptome profile and previously proposed phenotypes. While expression of iNOS, a major transcription factor for the pro-inflammatory M1 phenotype, was increased by HFD, gene enrichment analyses did not identify a broader M1-like transcriptome profile in macrophages from HFD-fed mice.

Consistent with Cochain’s description, Depuydt et al. identified three macrophage clusters (My.0, 1, 2) in human carotid endarterectomy samples, including two (My.0, My.1) that displayed pro-inflammatory profiles. The My.0 subset was preferentially involved in *Leukocyte transendothelial migration, Leukocyte extravasation signalling* (similarly to Cochain’s Inflammatory macrophage subset), and *Inflammasome activation* (*Il1B, Casp1, Casp4*), while My.1 macrophages were notably characterised by signalling associated with “antigen presentation” (as Cochain’s Resident-like cluster) and expressed higher levels of *Tnf* and *Tlr4*. Furthermore, My.0 and My.1, like the Inflammatory mouse macrophage subset, expressed anti-inflammatory genes (e.g., *Klf4*), suggestive of an inhibitory feedback loop in these pro-inflammatory macrophage populations. Depuydt’s last macrophage cluster, My.2, highly expressed *Trem2*, and, as Cochain’s TREM2^hi^ cluster, overexpressed lipid-metabolism-associated genes (*Abca1, Abcg1*) as well as *Cd9*, but mostly did not express pro-inflammatory genes. Instead, IPA indicated enrichment in metabolic pathways and liver X receptor/retinoid X receptor activation, consistent with a foam cell phenotype. Notably, My.2 cells also expressed *Acta2*, which encodes smooth muscle actin, together with macrophage markers (*Lgals3, Cd68*). However, the absence of transcription factors indicative of SMC lineage (e.g., *Myocd, Mrtfa*) suggested the My.2 cells were of myeloid origin and had gained characteristics of SMCs rather than being macrophages of SMC origin [10]. Thus, despite significant experimental differences (e.g., human vs. mice, aorta vs. carotid artery), Cochain’s and Depuydt’s findings are highly corroborative in their description of atherosclerotic plaque macrophages transcriptomic subsets and the subsets’ predicted functions. Specifically, Depuydt’s My.0 subset is highly reminiscent of Cochain’s Inflammatory cluster, the My.1 subset of Resident-Like macrophages, and the My.2 subset of TREM2^hi^ macrophages (Figure 1).

The association between a foam-cell phenotype and metabolic functions described by Depuydt was also observed by Kim et al. [7]. Following flow cytometry sorting of aortic macrophages (CD64+CD11b+CD45+) from *LdlR*^−/−^ animals on HFD, the authors demonstrated that foamy cells (positive fluorescent BODIPY stain) preferentially expressed genes associated with cholesterol transport (e.g., *Nr1h3, Abca1, Lipa*) and uptake (*Cd36, Mertk*), as well as efferocytosis (e.g., *Cd36, Mertk, Nr1h3*). Non-foamy cells (BODIPY-negative) expressed a number of pro-inflammatory (e.g., *I1b, Nfkbia, Tlr2, Tnf*), as well as some anti-inflammatory (e.g., *Gpr183, Tgfbr2*), genes. Thus, despite the lack of single-cell resolution, the conclusions from Kim’s foamy vs. non-foamy cell transcriptomic profiling are in line with the findings by Cochain and Depuydt, supporting (i) the coincidence of pro- and anti-inflammatory functions within both foamy and non-foamy cells, and (ii) the division of lipid metabolism (foamy) and pro-inflammatory (non-foamy) functions.

### T-Cells

3.2

Winkels’ scRNAseq analysis of aortas from 20-week-old female *ApoE*^−/−^ mice identified 5 clusters of T cells (*Cd3e*) [6], consistent with Th17 cells expressing *Rorc*, Th2 expressing *Gata3, Il-4* and *Il-13*, memory T-cells expressing *Sell* (CD62L) and *Il7r*, CD8^+^ cytotoxic T-cells (*Cd8a*) and a mixed population of *Cd4*^+^*/Cd8*^+^ T-cells. CyTOF analysis of these aortas supported these observations and identified 7 subsets of T-cells, including a γ/δ-T-cell subset and 2 CD8^+^ cytotoxic T cell subsets (Ly6C^+^ and Ly6C^−^), unidentified in the scR-NAseq analysis. While CyTOF analysis revealed that both T-helper subsets (Th17 and Th2) expressed folate receptor 4 (FR4), a marker of regulatory T cells (Tregs), these clusters lacked the Treg-defining CD25. Given that antigen-stimulated CD4^+^ T-cells up-regulate FR4 and down-regulate CD25, the authors conclude that most T-cell clusters were likely effector or central-memory, and not Tregs [6]. This is in line with previous reports of Treg disappearance in the plaque [23], potentially caused by conversion into effector T-cells [24].

Importantly, T-cell heterogeneity was greatly reduced in healthy (8-week-old mice), compared to atherosclerotic aortas from ApoE^−/−^ mice. A single cluster was identified and was transcriptionally similar to the memory T-cell cluster observed in atherosclerotic aortas. In line with this, and although the overall proportion of T-cells did not change, exacerbation of atherosclerosis with HFD led to a further reduction in the proportion of memory T-cells, as well as Th2 cells, while Th17 cells, overall described as pro-atherogenic [25], were increased. In addition to changes in the proportions of these T-cell clusters, HFD exposure induced marked changes in their transcriptional profiles. While gene enrichment analyses indicated Th2 cells were involved in cytokine and chemokine release, IPA revealed that HFD led to a reduction in Pigment Epithelium-Derived Factor (PEDF) signalling, accepted to be athero-protective [26], and apoptotic pathways towards more autophagy and hypoxia-associated signalling. In CD4^+^/CD8^+^ T-cells, memory T-cells, and CD8^+^ T-cells, a transcriptional switch from a recruitment phenotype in chow-diet animals (homing receptors, cellular motility genes) towards a pro-inflammatory phenotype (e.g., increased IFN-γ signalling) was observed in HFD animals.

Critically, deconvolution analysis of bulk RNA sequencing of human carotid plaques identified the same five T-cell subsets, with CD8+ T-cells being the most populous, followed by memory T-cells, Th17, CD4^+^/CD8^+^, and Th2 cells. In line with the observation made in mice that the proportion of memory T-cells decreased with the progression of atherosclerosis, Winkels et al. found that the proportion of memory T-cells, but no other leukocyte cluster, negatively predicted ischemic cardiovascular events in the same patients [6]. While this may suggest an overall protective function for memory T-cells, it is unclear whether their decline at the plaque site is a cause or consequence of plaque progression. While tissue-resident memory T-cells were identified within non-lymphoid organs and shown to persist and provide a tissue-restricted response following reactivation with their cognate antigen, it is not clear here whether the memory-like T-cells found in healthy aortas represent a type of tissue-resident T-cells.

Following scRNAseq analysis of all live cells in human carotid endarterectomy specimens, Depuydt’s unbiased clustering revealed 4 classes of T-cells (*Cd3e*+), mostly mixes of CD4^+^ and CD8^+^ T-cells. Reclustering of the CD4^+^ T-cells (*Cd4* expression > *Cd8* expression) identified five subgroups, which, somewhat surprisingly and in opposition to the findings by Winkels, differed mostly by their activation status and not by the markers used to define CD4^+^ T-helper subsets. [10]. Two subclasses were labelled as cytotoxic CD4^+^ cells (expressing *Gzma, Gzmk, Prf1*) and differed by their expression of *Cd28*. The CD4^+^CD28^−^ subset was confirmed by flow cytometry to produce more Granzyme B than the CD4^+^CD28^+^ subset and likely corresponds to CD4^+^CD28^null^ cells previously reported to correlate with increased risk of cardiovascular events [27]. Two other subclasses of CD4+ T-cells expressed *Il7r, Lef1* and *Sell*, thus suggesting the subsets contain naïve and central memory T-cells. The last subset expressed *Foxp3, Il2Ra* and *Ctla4*, and was thus labeled as Tregs. Interestingly, some of these cells co-expressed *Foxp*3 with the transcription factor *Rora*, associated with enhanced Treg immunosuppressive ability [28], or *Gata3*, which can prevent polarisation of Tregs towards other T-helper subsets [29]. Thus, Depuydt’s description of CD4^+^ T-cell plaque subpopulations highlighted that the main differences lay in the activation state of the cells, rather than in their alignment to classical T-helper subsets, as described by Winkels. In support of this, Depuydt et al. reported that expression of the T helper-specific transcription factors *Tbx21* (Th1), *Gata3* (Th2), and *Rorc* (Th17) was not associated with a specific cluster.

Reclustering of the CD8^+^ T cells (*Cd8* expression > *Cd4* expression) revealed three subgroups: an effector-memory subset (*Gzmk, Gzma, Cd69*), cytotoxic CD8^+^ T-cells (*Gzmb, Tbx21, Nkg7, Gnly, Znf683, Cx3cr1*, lack of *Cd69*), and central-memory CD8^+^ T cells (*Lef1, Sell, Il7r, Ltb*). Non-cytotoxic T cells expressed CD69, indicating a recent TCR activation, but the authors were not able to define the antigen specificity due to the lack of prior characterisation of potential atherosclerosis-specific antigens for CD8^+^ T-cells. Nonetheless, their observation suggests a separation of cytotoxic and antigen-presenting functions within CD8^+^ T-cells at the plaque site.

In a comparison of blood and plaque T-cells from carotid endarterectomy patients, Fernandez et al. [9] found that effector-memory cells (CCR7^low^CD45RA^low^) were enriched at the plaque site and displayed a resident-like phenotype (*Cd69*+ [30]). Of the four plaque-enriched T-cell subsets, all were effector-memory, three were CD8^+^ T-cells, and one corresponded to CD4^+^ T-cells. Of note, the CD4^+^ and 2 of CD8^+^ clusters displayed strong transcriptomic similarities, suggesting similar functions in atherosclerosis. One of these CD8^+^ clusters was exclusively detected in the plaque (as opposed to blood) and expressed CD103^+^, indicative of tissue-resident memory T-cells. The remaining plaque-enriched CD8^+^ cluster shared similarities with blood “effector memory cells re-expressing CD45RA” (EMRA) T-cells, suggesting a terminally differentiated phenotype.

By a combination of CyTOF and Cellular Indexing of Transcriptomes and Epitopes by Sequencing (Cite-Seq), plaque T-cells were found to be more activated (e.g., *Klrd1, Klrc1, Cxcr3, Stat3, Ifngr1, Hla-B, Ccl5, Ccl4, Cxcr6*), and exhausted (*Pd1, Eomes, Lag3*), as observed for exhausted T-cells in the tumour microenvironment [9], than their blood counterparts. Exhausted and activated T-cell subsets co-existed within the same plaque, suggesting that highly activated T-cells may undergo exhaustion, likely due to chronic unresolved local inflammation, thus leading to a progressive loss of T-cell function. The fact that PD-1 deficiency/blockade promotes atherosclerosis in vivo [31,32] and that cancer patients receiving PD-1 blockers as checkpoint inhibitors are at increased risk of atherosclerotic disease seems to suggest that T-cell exhaustion at the plaque site may be athero-protective [33].

ScRNAseq analysis of symptomatic (recent stroke or myocardiac infarction, MI), compared to asymptomatic, plaques revealed that CD4^+^ T-cells in asymptomatic patients displayed activation and effector functions (e.g., *IFNγ pathway*) and upregulated IL-1 and IL-6 pathways, while T-cells from symptomatic subjects did not activate IL-1 and IL-6 pathways, but upregulated pathways associated with T-cell migration (e.g., *RhoGTPase pathway*) and differentiation (e.g., *Wnt signalling* and *IL-2 signalling* pathways). Asymptomatic CD8^+^ cells also displayed increased IL-6 and TNFα signalling, and distinct genes and pathways suggested specific chemotactic (*Cxcr4, Ccl4l2*) and cytotoxic (*GzmB, Granzyme* pathway) functions in asymptomatic cells. Symptomatic CD8^+^ T-cells upregulated both inflammatory (e.g., *IL-8 and CXCR2 pathway, Positive regulation of cytokine-mediated pathways*) and exhaustion-associated pathways (e.g., *Cellular senescence, T cell exhaustion pathway*), again suggesting that exhaustion reprogramming at plaque-site may be initiated following increased/prolonged T-cell activation. However, because plaque samples were obtained after the occurrence of a cardiovascular event, it is unclear whether the differences between symptomatic and asymptomatic T-cells predated, and therefore may have promoted, the event, or developed consequently to the event. Notably, the evidence of increased T-cell exhaustion in the plaques of symptomatic patients is in apparent contradiction with the hypothesis that T-cell exhaustion may protect against atherosclerosis progression [33], and suggests that T-cell exhaustion is a consequence, rather than a cause of, plaque rupture, and could be due to the heightened inflammatory activity that follows a cardiovascular event [34] (Figure 2).

### B-Cells

3.3

As opposed to macrophages and T-cells, deconvolution of bulk RNAseq from micro-dissected aortic tissues revealed that B-cell populations were preferentially located in the ALTO-rich adventitia of atherosclerotic aortas from *ApoE*^−/−^ mice [6]. While unsupervised clustering of all leukocytes identified two classes of B cells, within *Cd19*^+^ events, three subsets were identified, which differed by the expression of *Spn* (Sialophorin, CD43) and of the CD45 isoform *b220* [6]. CyTOF analysis of atherosclerotic aortas also identified these three subsets, namely B220^high^CD43^low^ (Cluster 1), B220^high^CD43^neg^ (Cluster 2) and B220^neg^CD43^high^ (Cluster 3). Thus, these clusters matched the core marker profile of either athero-protective B1-cells (B220^low^CD11b^+^CD23^−^, Cluster 3) or pro-atherogenic B2-cells (B220^+^CD23^+^CD11b^−^, Clusters 1 and 2) [29]. In line with their proposed anti-atherogenic functions, B1-like cells were found to decrease in aortas from animals exposed to HFD, as opposed to a chow diet, while pro-atherogenic B2-like cells were increased [6]. However, while IPA of Differentially Expressed Genes (DEG) revealed that clusters 1 and 2 upregulated *Antigen presentation, co-stimulation, Antibody generation, Cell adhesion* (Cluster 1), and *Cell division* (Cluster 2); Cluster 3 up-regulated pro-inflammatory TNF-signalling genes, in apparent opposition to the reported athero-protective effect of B1-cells. In line with increased TNF signalling in Cluster 3, flow cytometry analysis of CD43^high^B220^neg^ cells revealed their ability to produce CCL5 (RANTES) upon stimulation, while B2-like cells (B220^high^) secreted pro-atherogenic IFN-γ and GM-CSF.

In a model of hyperhomocysteinemia (HHcy)-accelerated atherosclerosis in *ApoE*^−/−^ mice, B cells (*Cd79a, C79b, Cd19*) were found to be the most abundant leukocyte population (47.2%), together with macrophages [12]. Unsupervised clustering of *Cd45*^*+*^ cells identified 4 clusters of B-cells: Cluster 0 expressing *H2-Oa* and *Cr2d*, suggesting a role in antigen presentation; Cluster 8 expressing *Spib and Ms4a1*, indicative of memory B-cells; Cluster 12 expressing *Ssr4 and Xbp1*, suggesting plasma cells. Intriguingly, the second most abundant B-cell cluster, Cluster 3, displayed no clear classic B-cell subtype profile. The authors compared the effect of HHcy, which drives atherosclerosis through endothelial injury, with that of HFD, which promotes LDL elevation in blood. They found that B-cell proportions were increased following HHcy (47.2%) compared to control *ApoE*^−/−^ mice (37.2%), while HFD intervention led to a reduction in B-cell proportions (19.9%), probably due to an increase in monocyte/macrophages in these mice. Specifically, there was a sharp increase in antigen-presenting B-cells (Cluster 0) in response to HHcy (30.2% vs. 13.6%), but not in response to HFD (8.13%).

To better characterise B-cell heterogeneity, Ma et al. performed a re-clustering analysis focusing on B-cells alone, which identified 8 B-cell subsets. Among them, the proportion of the most abundant cluster in *ApoE*^−/−^ aortas was dramatically increased following HHcy induction (67.50% vs. 30.73%), while the increase in response to HFD was modest (36.1%). GO pathway enrichment analyses indicated that the dominant pathway in this B-cell cluster related to *Antigen presentation*, exemplified by the upregulation, compared to other B-cell subsets, of MHCII-related genes, such as *H2-Oa, H2-Eb2, H2-DMa, Cd74* and *Ciita*. The expression of chemokine receptors *Ccr6* and *Cxcr4* was also elevated in this cluster, suggesting potential migration from peripheral sites as a possible source of the B-cells in this cluster.

Having observed an increase in CD4^+^ T-cells in HHcy mice, Ma et al. conducted cell–cell interaction analysis, which revealed B-cells as a major source of MHCII signalling, with T cells being the prominent receiver of this signalling. Furthermore, B-cells were the dominant source of MHCII-signalling in HHcy *ApoE*^−/−^ mice specifically, while macrophages and DCs were the biggest contributors to antigen presentation in *ApoE*^−/−^ and *ApoE*^−/−^+HFD animals. In addition to higher B-cell proportions in HHcy *ApoE*^−/−^ mice, the authors speculate that the increased expression of MHCII-associated molecules in HHcy *ApoE*^−/−^ mice likely promotes antigen-presentation signalling between B and CD4^+^ T-cells in this model. In line with the findings by Winkels et al., colocalisation between B-cells and T-cells mostly occurred in adventitial ATLOs, suggesting these as preferred sites of antigen presentation by B-cells to T-cells [12].

Mechanistically, in vitro exposure of splenic B-cells to Hcy upregulated MHCII expression at both the gene and protein levels and augmented the cells’ ability to activate T-cells in a co-culture model [12]. Interestingly, the antigen-presentation capabilities of DCs and macrophages were mostly unaffected by Hcy.

CD11c+ B cells are a newly described subset of B-cells, shown to expand with age, that have been associated with autoimmune diseases. Pattarabanjird et al. found that CD11c+ B-cells were increased in the spleen, blood and bone marrow of 100-week-old, compared to 50-week-old *ApoE*^−/−^ mice, and the proportion of CD11c+ B cells in these compartments, but not that of total B-cells, correlated with plaque burden in aortas [15]. Flow cytometry analysis of PBMCs from patients with Coronary Artery Disease (CAD) revealed that CD11c^+^ B-cell proportions were increased in patients with high, compared to low, CAD severity [15]. CITE-Seq of these PBMCs (50 proteins, 488 genes) identified four CD11c^+^ B-cell clusters, based on the expression of CD27 and IgD. These were named double negative 2 (DN2, CD11c^+^CD27^−^IgD^−^, also CXCR5^−^), unswitched memory (CD11c^+^CD27^+^IgD^+^, also IgM^−^), activated naïve (CD11c^+^CD27^−^IgD^+^), and age-associated B cells (ABC, CD11c^+^CD27^lo^IgD^−^, also CXCR5^−^). This subtyping allowed the identification of the specific CD11c^+^ B-cell subtypes associated with high CAD severity, namely the DN2 and ABC, but not the unswitched memory or activated naïve subsets. IPA of DEG in DN2 cells revealed enrichment in the inflammatory *Toll-like receptor* (*TLR*) *and Interferon-γ* pathway, as well as *Autophagy*. In ABC CD11c+ B cells, DEG and IPA suggested that BCR signalling may be enriched in cells from patients with severe atherosclerosis, as exemplified by higher expression of *Itgax* (CD11c), a marker of activation, and lower expression of *Cd72*, an inhibitor of BCR signalling [15].

CD11c+ B cells notably mediate inflammation via autoantibody production, and malondialdehyde low-density lipoprotein (MDA-LDL) is a known oxidation-specific epitope deposited in atherosclerotic plaque [15]. While IgMs to MDA-LDL are thought to be athero-protective, IgGs to MDA-LDL likely promote disease, and Pattarabanjird et al. found that plasma levels of IgG were correlated with the proportion of the DN2 CD11c^+^ B-cell subset [15]. Furthermore, B-cell-focused re-clustering analysis of a previously published scRNA-seq dataset from fresh coronary atheroma samples identified 6 clusters of B cells, notably including 2 subsets of CD11c^+^ B-cells, corresponding to the previously identified DN2 and activated naïve subsets, suggesting that these cells can infiltrate the coronary atheroma. Taken together, these results suggest that increased circulating proportions of DN2 CD11^+^ B-cells may drive CAD severity by mechanisms including proliferation and the production of atherogenic IgGs to MDA, either in the blood or at the atherogenic plaque site, which could be driven by autophagy and augmented TLR signalling [15].

Of note, and in opposition to the studies by Ma and Pattarabanjird, Depuydt et al. only identified a single small and homogenous cluster of B cells (*Cd79a, Fcer2, Cd22, Cd79B*) in human carotid endarterectomy samples [10]. While a number of reasons may underly this apparent discrepancy, it is likely that the lack of focus on immune (CD45^+^) cells and the absence of a B-cell-centred clustering analysis in Depuydt’s study will have significantly contributed to a lower sensitivity in the detection of the different subsets of B-cells populating the human atherosclerotic plaque.

### Non-Immune Cells

3.4

#### Endothelial Cells

3.4.1

Depuydt’s unsupervised clustering analysis of total scRNA seq from carotid endarterectomy samples identified 2 clusters of endothelial cells (*Cd34*^+^), and CD34^+^-focused re-clustering revealed 4 subclusters [10], although the proportion of each cluster is unclear. E.0, E.1, and E.2 displayed the classical endothelial markers *Cd34* and *Pecam1*, together with the vascular endothelial marker *Tie1*. The E.0 cluster differed by its expression of *Ackr1*, suggesting a *vasa vasorum* origin, and *Prcp*, which promotes EC proliferation in angiogenesis and endothelium repair [35]. Clusters E.1 and E.2 expressed *Vcam1*, involved in adhesion and transmigration of leukocytes, indicating an activated endothelium. E.1 expressed *Bmp4*, along with other extracellular matrix genes, while E.2 was characterised by mobility genes, notably *Fgf18* and *Heg1*. The last EC subcluster, E.3, expressed *Cd34* and *Pecam1* together with SMC markers (e.g., *Acta2, Notch3, Myh11*), suggesting a subset of EC undergoing endothelial-to-mesenchymal (EndMT) transition or cells of SMC origin undergoing mesenchymal-to-epithelial transition (MTEnd).

The findings by Depuydt were echoed in a study by Zhao et al., who identified 8 EC (*Cdh5*^+^
*Pecam1*^+^) clusters in aortas and hearts from *LdlR*^−/−^ mice on a diabetogenic diet (high fat and high sugar) [11]. The most abundant two clusters (EC_1, 32%; EC_2, 26%) highly expressed genes involved in angiogenesis (*Flt1, Kdr*), reminiscent of Depuydt’s E.0 cluster, and fatty acid metabolism (*Fabp4, Cd36, MgII*). GO pathway analysis identified *Vasculature Development* as the top-enriched pathway for both clusters. Interestingly, EC_2 exhibited an athero-protective phenotype (high *Klf2* and *Klf4*, enrichment in the *Regulation of cellular response to stress* pathway), and this cluster was increased after a diabetogenic diet, suggesting EC_2’s involvement in a negative feedback loop during atherosclerosis plaque development. EC_3 (24%) and EC_7 (4.9%) displayed a profile aligning with a primary function in maintaining the integrity of the endothelial barrier, and both clusters were dramatically reduced following a diabetogenic diet (EC_3, chow: 43%, diabetogenic: 1.5%; EC_7, chow: 9%, diabetogenic: undetected). EC_4 (12%) appeared to be restricted to the heart tissue (*Fabp3, Myl2, Myl3*) and thus did not have an equivalent in Depuydt’s study. EC_5 (3.6%), EC_6 (3.3%) and EC_8 (2.5%) expressed both endothelium-associated genes (e.g., *Pecam1, Cdh5, Vwf*) and mesenchymal genes (*Dcn, Mgp, Eln, Fn1, Vim, Tgfbr2*), suggesting that these cells are undergoing EndMT, as observed for Depuydt’s E.3 cluster. In support of an EndMT process, GO pathway analysis of these EC clusters revealed enrichment in *Vascular development* (all clusters), *Extracelluar matrix structural constituent* (EC_5), *Collagen-containing extracellular matrix* (EC_6) and *Tissue remodeling and mesenchyme development* (EC_8).

To characterise the role of EndMT-undergoing cells, Zhao et al. transcriptomically compared ECs that co-expressed mesenchymal markers with ECs that did not. GO pathway analysis revealed that EndMT cells were preferentially upregulating proatherogenic pathways, including *ECM organisation, A*poptosis, the pro-inflammatory *NF-κB pathway* (*Nfkbiz*) and *Adhesion molecule binding* (*Vcam1, Icam1, Cxcl16*) [11]. Furthermore, diabetogenic-diet feeding further increased both the frequency of EndMT ECs (12.6% vs. 6.6%) and their expression of several genes in the pro-atherogenic pathways above (e.g., *Ctsb, Ctsz, Bmp4, Tgfbr2, Vcam1, Icam1, Cxcl16, Nfkbiz*). Taken together, these results suggest the existence of a positive feedback loop: EndMT ECs drive atherosclerosis development, which in turn leads to more EndMT and to a more inflammatory EndMT EC profile [11] (Figure 3). GWAS trait enrichment at single-cell resolution identified EndMT ECs as the most enriched EC cluster in CAD traits [14], supporting their role as drivers of atherosclerosis progression.

#### Smooth Muscle Cells

3.4.2

The study by Depuydt et al. also identified a cluster of SMCs (*Myh11, Pdgfrb, Notch3, Mfap4*) within the non-immune clusters. Re-clustering revealed 2 subclusters of SMCs, a dominating cluster with a synthetic profile (*Mgp, Col3a1*) and one with a contractile profile (*Myh11, Acta2, Tagln*). A portion of the synthetic SMC cluster strongly upregulated genes associated with extracellular matrix (ECM) biology and decreased typical SMC markers, suggesting a location in the established cap of the plaque. Of these cells, some expressed *Klf4*, potentially indicating a transition to a macrophage-like phenotype.

SMC plasticity was the focus of a meta-analysis by Mosquera et al., which integrated scRNAseq human data from over 20 libraries and utilised markers of differential SMC phenotypes identified in mouse studies [14]. The authors describe a progressive loss of the SMC contractile profile with atherosclerosis development, together with a gain in the pro-inflammatory gene *Lgals3*^+^ (galectin 3), shared with macrophages and associated with disease progression, supporting a transitional SMC state. Transitional SMCs, which displayed a synthetic profile, were enriched in ECM-associated genes, in line with the observation by Depuydt. In addition to contractile and transitional SMCs, 3 other SMC subsets were identified, namely fibromyocyte, fibrochondrocyte (*Cytl1, Comp, Col1a1/2*), and foam-cell-like (*Apoe, Apoc1, Agt*) profiles were identified [14]. The disease relevance of SMC phenotypes was analysed using genetic prioritisation models and GWAS summary statistics; fibromyocytes were the most highly associated with CAD/Myocardial Infarction, while fibrochondrocytes were, expectably, enriched for coronary artery calcification [14]. These findings not only support the critical role of SMCs in driving vascular disease but also highlight how SMC plasticity underlies differential pathophysiological processes.

#### Regional Segregation of Transcriptomic Profiles

3.4.3

ScRNAseq of severe carotid plaque (type VII, calcified) from human carotid endarterectomy samples revealed the importance of cellular location on transcriptomic profile, predicted functions and trans-differentiation processes [13]. ScRNAseq of the calcified core part of the plaque (AC) and the non-atherosclerotic proximally adjacent (PA) region identified 3 subpopulations of SMCs, which appeared to form a continuum of cellular states. The starting SMC population was exclusively present at the PA region, and upregulated genes were involved in *Inflammatory responses* and *TNFα signalling*. Notably, C*3*, a complement component involved in opsonization and phagocyte recruitment, was highly differentially expressed by these cells, suggesting a major role of SMCs adjacent to the lesion in driving complement activation in atherosclerosis. The PA subsets of SMCs also displayed early migratory and trans-differentiation capabilities, exemplified by the expression of *Fbn1, Sema3c, Htra3*, and *C1qtnf3*. At the other end of the SMC continuum was a subset composed mostly (70%) of AC-residing cells, consistent with a synthetic osteoblast (*Postn*^+^)- and chondrocyte (*Crtac1*+)-like profile, which likely drives calcification (*Spp1, Ibsp, Tnfrsf11b*) and cartilaginous matrix-secretion (*Col4a1, Col4a2*). Somewhat unexpectedly, this subset of AC SMCs strongly upregulated *Sfrp5*, which codes for an anti-inflammatory adipokine, an inhibitor of Wnt signalling, thought to protect against atherosclerosis [36]. This suggests a reduction in SMC-driven inflammatory processes in the core, compared to the border, of the atherosclerotic plaque, and a shift towards matrix remodelling and calcification [13].

In contrast to the PA-to-AC continuum observed for SMCs, two distinctly separate clusters of ECs were observed in PA vs. AC regions. The first difference lay in the significantly lower number of AC ECs than PA ECs (488 vs. 2316), likely due to erosion and loss of endothelium integrity at the advanced lesion site. It is also possible that EndMT, mentioned above to be prevalent at the lesion, causes a subpopulation of ECs to lose common EC markers. As observed for PA SMCs, DEG in PA ECs support a pro-inflammatory profile, with the top 5 upregulated genes including *Il6, Ackr1*, which facilitates leukocyte recruitment, and *Hla-Dqa1* (MHC class II), which may support the activation of T-cells [13]. The upregulated genes in AC ECs suggest a role in intimal repair, characterised notably by the expression of *Itln*, coding for Omentin, which is thought to be athero-protective via inhibition of TNF-induced vascular inflammation and promotion of revascularisation. *Dkk2* encodes a Wnt antagonist that stimulates angiogenesis, and its up-regulation in AC ECs further supports a role in intimal repair [37]. Of note, the AC EC population upregulated *Fn1* (fibronectin), indicating that cell trans-differentiation, in this case EndMT, preferentially occurs at the core lesion site compared to neighbouring regions (Figure 3).

Another vascular region where non-immune cells were shown to play a critical role in atherosclerosis development is the arterial adventitia. Analysis of the aortic adventitia of *ApoE*^−/−^ mice identified six non-immune cell clusters that orchestrate leukocyte infiltration into the atherosclerotic plaque [8]. These clusters were defined as (from most to least populous): adventitial ECs (*Pecam1, Cldn5, Icam2, Esam, Tm4sf1, Egfl7*), SMCs (*Myh11, Flna, Tpm2, Acta2*), Mesenchymal (Mesen) cluster III (*Lpl*), Mesen IV (*Prelp*, cartilage marker [38]), *Ptn* heparin-binding cytokine [39])), Mesen I (*Mfap4* [40] and *Adamtsl2* [41] ECM proteins) and Mesen II (*Ly6a*, marker of stem cells [42]). Mesen II highly expressed *Ccl2, Pla1a* and *Pi16*, indicating an inflammatory state [43-45]. Cell–cell interaction analyses highlighted that mesenchymal cells interacted with myeloid cells, as well as with other mesenchymal cells. Notably, Mesen II cells expressing *Cd34* and *Cav1* interacted with *Sell* and *Icam1* on inflammatory macrophages, and both Mesen II cells and resident macrophages interacted with inflammatory cells through the Ccl2-Ccr2 and Cccl7-Cccr7 pathways. As these observations were made in *ApoE*^−/−^ mice on a normal chow diet, these observations could support the involvement of resident macrophages and mesenchymal cells in the early developmental stages of atherosclerosis [8].

## Discussion

4

Following its global development in the 2010s, a boom in the use of scRNAseq has led to a previously inconceivable wealth of data being generated in relation to the mechanisms of health and disease. The field of atherosclerotic research has been no exception. A large number of studies, combining omics techniques such as scRNAseq, CITE-Seq and CyTOF (Table 1, non-exhaustive), have drastically increased our knowledge of the cellular composition of the atheroma and our understanding of the cellular mechanisms driving disease progression. Beyond the identification of plaque cell populations and subsets and the description of their transcriptomic profile, the use of pathway analysis, cell mapping and ligand: receptor interaction technologies have generated critical information about the origin of cells at the plaque, their (trans)differentiation pathways and likely roles as senders and receivers of pro and anti-atherogenic signals (not covered in this review, see [8-10,12]).

Taken together, and despite significant apparent discrepancies, the studies reviewed here have substantially advanced our understanding of the pathophysiology of atherosclerosis. Notably, they have highlighted the extent to which cellular plasticity contributes to disease progression, which may pave the way towards therapeutic strategies to prevent or promote specific cellular phenotypic switches or trans-differentiation processes. Furthermore, clustering based on full transcriptomic profiles, as opposed to classical protein markers, has led to the realisation that cell subtypes previously thought of as distinct could perform very similar functions, such as CD4+ T cell clustering as cytotoxic, memory, or regulatory, irrespective of their expression of classical Th1, Th2, and Th17 markers [10]. In addition, several of these studies reported a separation of classical functions within a cell type. Indeed, pro-inflammatory cytokine production, antigen presentation, lipid handling and efferocytosis were associated with different plaque macrophage subsets [5,7,10], while pro-inflammatory responses and ECM remodelling were separated in EC and SMC clusters [13]. Thus, the inhibition or promotion of detrimental or beneficial functions (respectively) will most likely need to be achieved by specific targeting of the signalling pathways associated with the function of interest, rather than by the general targeting of the cell type classically associated with that function.

However, despite the number of published studies and the considerable bulk of data generated, consensus on issues that could be perceived as being of little complexity is still lacking, such as the predominant leukocyte population at the atherosclerotic lesion site and the relative proportions of others. While macrophages and T-cells mostly appear as the 2 major populations, it remains unclear which is the most prominent across at least 4 of the studies discussed here [5,6,9,10], with a 5th indicating B-cells as the leading population [12]. While some of these discrepancies may reflect genuine differences between species (man and mouse), models of atherosclerosis (*ApoE*^−/−^ vs. *LdlR*^−/−^ mice, HFD vs. HHcy), or sexes (studies in male or female mice only), these obvious confounding factors cannot entirely explain the lack of consensus. It is notable, for example, that Winkels identified T-cells as the most prominent leukocyte population in the atherosclerotic aortas of *ApoE*^−/−^ mice by scRNAseq, in opposition to CyTOF analysis of artoas from the same model [18], while deconvolution of bulk RNAseq from whole aortas revealed macrophages as the most prominent leukocyte population [6]. A similar discrepancy was observed when human carotid endarterectomy samples were analysed by CyTOF (T-cells predominant) or deconvolution of bulk RNAseq (macrophages predominant). A major drawback of single-cell, as opposed to bulk tissue, omics analyses is the requirement for the preparation of a viable single-cell suspension, which can be difficult to optimise, especially for hard-to-digest tissues, such as arteries [13]. It is likely that tissue digestion will preferentially release some cell types over others, notably depending on their location within the arterial wall (i.e., intima, media, or adventitia), which may explain the inconsistencies reported by Winkels et al. between their single-cell techniques (scRNAseq or CyTOF) and deconvolution of bulk tissue analyses. B-cells, notably, were found to reside mostly in harder-to-digest adventitia [6,12] and may therefore be underrepresented in single-cell preparations of whole aortas.

When discrepancies are observed between studies using the same model and single-cell technique, for example, between Winkels and Cochain’s scRNAseq analysis of aortas from *LdlR*^−/−^ mice on HFD, differences in the digestion protocol may be considered. While most studies have used a mix of the same three enzyme groups (collagenases, hyaluronidases and DNases) and performed digestion over about 1 h, differences in the exact enzyme types, their concentration, the presence or absence of serum, the starting sample size, may all contribute to differential and preferential population release or viability. When digestion is followed by live cell sorting (e.g., CD45^+^), sorting time may also affect final live leukocyte proportions, as the viability of different populations will be differentially affected by the sorting step. Going forward, it is likely that the development of new single-cell in situ technologies, which allow spatial transcriptomic analyses in tissues, will remove the need for single-cell preparations and therefore may support more reproducible findings.

It is noteworthy, however, that despite some apparent discrepancies in the bigger picture of atherosclerotic plaque composition, significant points of agreement on finer subset descriptions have emerged between studies, sometimes across species and models. For example, despite a disagreement regarding macrophage overall predominance at the lesion site, Cochain, in murine aortas, and Depuydt, in human carotids, identified 3 macrophage plaque subsets sharing remarkably similar transcription profiles and predicted functions (Figure 1). Similarly, independent analysis of EC subpopulations in human carotid (Depuydt) and mouse aortic (Zhao) plaque, and despite discrepancies in the number of clusters identified (4 vs. 8, respectively), described their most populous cluster(s) as being involved in angiogenesis, while their least populous clusters were showing markers of transition towards an SMC phenotype, together with predicted functions in ECM remodelling. Such striking similarities, when observed across studies, species, models and methods of analysis, likely highlight key cellular processes and critical cellular functions in disease development and progression.

An interesting, shared observation between the studies by Cochain and Winkels is the increased proportions of T-cells in atherosclerotic aortas from *ApoE*^−/−^ vs. *LdlR*^−/−^ mice on HFD. While atherogenesis is promoted, in both models, by non-HDL hyperlipidaemia, different lipid profiles underlie disease progression. Notably, elevated VLDL plasma levels were identified as major drivers of plaque growth in *LdlR*^−/−^ mice, while in the *ApoE*^−/−^ model, reduced HDL levels were identified as critical [19]. Therefore, it is possible that a patient’s blood lipid profile will impact plaque cellular composition and may thus determine the most promising cell target for specific therapeutic intervention. Of note, comorbidities that drive CV risk, such as chronic kidney disease or diabetes, may be associated with dyslipidemia [46] and may therefore promote changes in cellular composition and functions within the atherosclerotic plaque. Other patient-specific factors, notably those associated with genetic susceptibility to CVD, can drive differences in plaque composition and driving mechanisms. Turner et al. incorporated GWAS data with plaque scRNA-seq data to map CAD susceptibility loci to specific lesion cell populations, which may help to identify patient-specific targets and cell types for therapeutic intervention [47].

Another shared conclusion between several of the studies discussed here is the added value of obtaining information about cell location within the plaque. By using deconvoluted bulk transcriptomic data from micro-dissected lesional intimal, medial and adventitial tissues, Winkels et al. demonstrated that B-cells were mostly located in adventitial ATLOs, where Ma et al. demonstrated that most B:T-cells interactions, specifically antigen presentation by B-cells to CD4^+^ T-cells, occur. By comparing EC and SMC transcriptomic profiles at the core (AC) and immediately next to the (PA) lesion site, Alsaigh et al. demonstrated that inflammatory responses are mostly driven by ECs and SMCs adjacent to the plaque, while at the lesion core, these cells lose their inflammatory profile in favor of functions in matrix remodelling and calcification. Of note, the AC region was the main site of cell trans-differentiation, including EndMT and differentiation of SMC into osteoblast-like cells. Several studies have also used in situ spatial transcriptomic techniques, although not at the single-cell level, to elucidate how location within the plaque dictates cellular transcriptomic profiles and functions. In line with Alsaigh’s findings, proximal coronary artery plaque regions were found to express more inflammatory pathways and fewer ECM pathways than the fibrotic core, but distance to the core was not the only determinant of transcriptomic signatures, with position with respect to blood flow having a critical differential effect [48]. While the proximal region (facing blood flow) overexpressed pathways associated with thrombosis, the distal region (away from blood flow) strongly upregulated the gene coding for the macrophage-associated MMP9, which was found associated with the future risk of CV events [48]. In another study, in situ spatial transcriptomics identified two main endothelial transcriptomic profiles, which were mutually exclusive and located in the adventitia or luminal endothelium [49]. Thus, information about cell location within the complex environment that is the atherosclerotic plaque may be key to further unravelling the role of the various cellular subsets identified by single-cell omics and may help to reconcile some of the apparent discrepancies highlighted here. The emergence of high-resolution spatial transcriptomics technologies, such as hybridization-based in situ sequencing (HybISS) and fluorescence in situ hybridization (FISH), will likely provide the next leap in our understanding of the pathophysiology of atherosclerosis.

Importantly, while transcriptional changes drive cellular functional changes during atherosclerosis progression, epigenetics controls these transcriptomic changes through mechanisms including histone modification and DNA methylation [50,51]. While the review of the epigenetic mechanisms controlling atherosclerosis progression is beyond the scope of this study, a better understanding of the “writers”, “readers”, and “erasers” of epigenetic alterations within different plaque cell populations and subsets may highlight novel targets for epigenetic therapies.

Thus, in the near future, spatial transcriptomics and epigenetic analyses, performed with single-cell resolution, will likely drive the development of next-generation patient-focused, cell-specific, therapeutic strategies against the world no. 1 killer.

## Figures and Tables

**Figure 1 F1:**
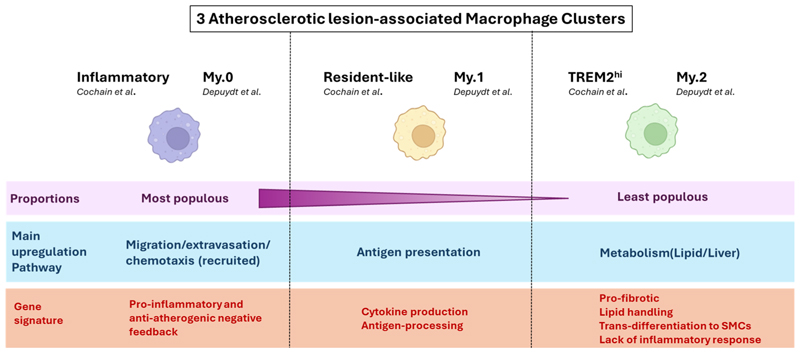
Notable points of consensus in atherosclerotic lesional macrophage clustering between mouse (Cochain) and human (Depuydt) plaques. A summary of converging conclusions regarding macrophage transcriptomic and functional heterogeneity at the atherosclerotic lesion, between scRNAseq analysis of mouse (*LdlR*^−/−^ mice on HFD) and human (Carotid endartectomy patients) plaque [5,10].

**Figure 2 F2:**
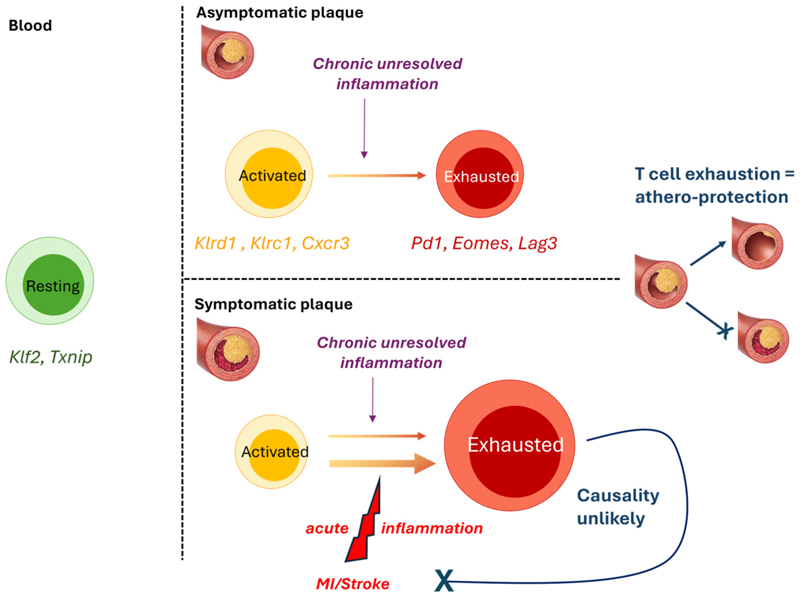
T-cell exhaustion at the plaque site is likely a consequence, rather than a cause, of increased inflammation and disease progression. Schematic representation of the findings by Fernandez et al. showing that plaque T cells are more activated (yellow) than their blood counterparts (green) and display signs of exhaustion (red), suggesting that exposure to chronic inflammation at the plaque may drive exhaustion reprogramming in activated T-cells [9]. Given the prior observations that T-cell exhaustion is athero-protective [27-29], it is unlikely that increased T-cell exhaustion in symptomatic plaque is a driver of disease progression. Instead, acute systemic inflammation as a result of MI or stroke likely contributed to further inflammation and T-cell exhaustion at the plaque site. Specifically elevated genes indicative of resting (green), activated (yellow), and exhausted (red) T-cells are indicated.

**Figure 3 F3:**
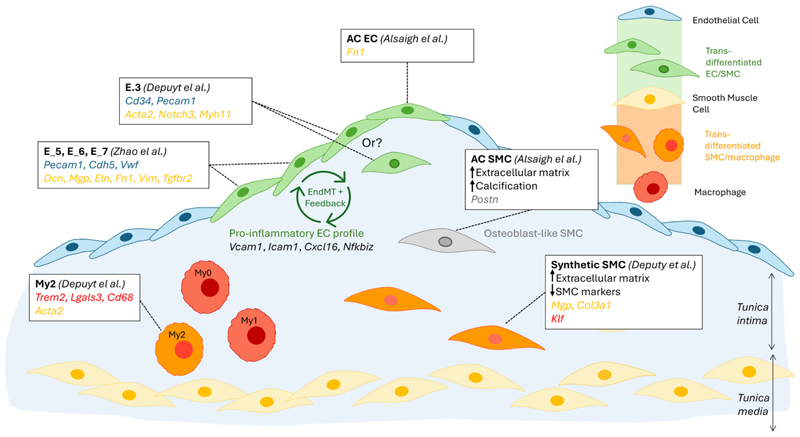
Schematic summary of cellular trans-differentiation at the atherosclerotic lesion site. The differentiation of plaque macrophages into smooth muscle cell (SMC)-like cells (orange); plaque SMCs into macrophage-like cells (orange), osteoblast-like cells (grey), and potentially endothelial cells (EC)-like cells (green); and plaque ECs into SMC-like cells (green) has been supported by sRNAseq findings. The supporting publications, specific cell subsets affected, and cell-type-specific marker genes used are indicated in the framed text box. Gene markers are colour-coded as follows: blue, EC, yellow, SMC, red, macrophage, grey, osteoblast. A potential positive feedback loop, suggested by Zhao et al., in which EndMT drives an inflammatory profile in ECs, which in turn drives EndMT, is also indicated [10,11,13].

**Table 1 T1:** Summary of findings from single-cell omics analyses of atherosclerosis in chronological order of publication.

1st Author/Date	Technique	Model	Main Findings
Cochain [5] June 2018	scRNAseq of immune (CD45+-enriched) cells from mouse aorta	LdlR^−/−^ and ApoE^−/−^ male mice on chow diet vs. high-fat diet (HFD) 11 weeks vs. 20 weeks	Macrophages are the predominant leukocyte population in plaque and are highly heterogeneous in disease. Macrophage populations specific to atherosclerosis are present at different stages of the disease.
Winkels [6] June 2018	CyTOF and scRNAseq on immune (CD45+-enriched) cells from mouse aorta Deconvolution of bulk RNAseq	LdlR^−/−^ or ApoE^−/−^ female mice on chow diet vs. HFD (12 weeks) Human carotid plaques	LDLR^−/−^ or ApoE^−/−^ mice have different atherosclerotic leukocyte populations and distributions. Both mouse and human carotid plaques are enriched in T-cells, macrophages, and monocytes. Each of these populations is highly transcriptomically heterogeneous.
Kim [7] October 2018	scRNAseq of immune (CD45+-enriched) cells from mouse aorta Bulk RNAseq of flow-sorted foamy vs. non-foamy macrophages	LdlR^−/−^ mice on chow diet vs. HFD (12 weeks)	Separation of functions between non-foamy and foamy macrophages. Non-foamy cells were mostly involved in (pro- and anti-) inflammatory responses, while foamy cells upregulated pathways involved in lipid metabolism. Foamy cell proportions correlated with atherosclerosis severity.
Gu [8] June 2019	scRNAseq on total cells from mouse aortic adventitita	ApoE^−/−^ vs. wildtype male mice (12 wo) on chow diet	Non-immune cell populations in aortic adventitia are numerous, and heterogeneous, and they contribute to the recruitment of immune cells in the early stages of atherosclerosis.
Fernandez [9] October 2019	CyTOF, CITE-Seq, and scRNAseq immune (CD45+-enriched) cells from carotid artery plaques of patients	Patients with clinically symptomatic atherosclerotic disease compared to those with asymptomatic disease	T-cells are the most abundant population of plaque leukocytes, and effector memory T-cells are specifically enriched in plaques compared to blood. T-cells at the lesion site display features of activation and immune exhaustion, indicating a likely transition from activation to exhaustion. T-cell exhaustion is further increased in symptomatic plaques.
Depuydt [10] September 2020	scRNAseq of total cells from human carotid endarterectomy tissue	Patients with carotid artery disease	Described the cellular landscape of human atherosclerotic plaque. Revealed high heterogeneity and plasticity of T cells, macrophages, and endothelial cells. Identified likely cell-cell interactions.
Zhao [11] May 2021	scRNAseq of EC (CD31+)-enriched cells from mouse heart and aorta	LdlR^−/−^ mice on chow vs. diabetogenic diet (12 weeks)	High EC heterogeneity in healthy and atherosclerotic aortas. EndMT drives atherosclerosis and is increased by it.
Ma [12] July 2022	scRNAseq of immune (CD45+-enriched) cells from mouse aorta	ApoE^−/−^ mice fed with homocysteine vs. HFD	The majority of B-cells in atherosclerotic aortas function as antigen-presenting cells, and this is exacerbated following homocysteine, but not HFD, acceleration. Antigen-presenting B-cells accelerate atherosclerosis by interacting with CD4+ T cells.
Alsaigh [13] October 2022	scRNAseq on the total aortic cell population. EC and SMC-focused analysis	Patients with asymptomatic type VII plaque, comparing composition at calcified atherosclerotic core and proximal adjacent region	VSMCs and ECs contribute to calcification in atherosclerotic plaque through different pathways. Transcriptomic profiles of EC/SMC situated at the core of the plaque are different from those at the border and suggest that cells at the core drive plaque remodelling and calcification, while cells at the border drive inflammation.
Mosquera [14] November 2023	Integrated meta-analysis of 22 scRNA-seq libraries coupled with GWAS summary statistics and genetic prioritisation.	Human coronary arteries, atherosclerotic lesions, and non-lesions	SMCs are highly plastic at the plaque site. Different SMC phenotypes likely drive different pathologies. Fibromyocytes are highly associated with CAD/Myocardial Infarction, while fibrochondrocytes are enriched for calcification.
Pattarabanjird [15] January 2024	Flow cytometry on blood, spleen, and bone marrow from ApoE^−/−^ mice. CITEseq on blood from coronary artery disease patients	Ageing Apoe^−/−^ mice, 50 vs. 100-week-old Patients with coronary artery disease (CAD), high vs. low severity	The proportion of CD11c+ B cells increased in blood, spleen, and bone marrow from older mice and correlates with the increase in aortic plaque. Specific CD11c+ B-cell subsets (age-associated and double negative 2, DN2) were increased in blood from patients with high severity CAD, compared to low severity. DN2 CD11c+ B-cells can be recruited from the blood to the plaque.

**Table 2 T2:** Comparison of circulating and plaque proportions of major leukocyte populations in humans and mice. macrophages

	MOUSE	HUMAN
	Blood[20]	Plaque [Table-fn TFN1][5,6,18]	Blood[20]	Plaque [Table-fn TFN1][6,9,10]
Monocytes/macrophages	1.4–2%	22–60%	2.3–4.6%	13–65%
T-cells	7–9%	28.3–49%	11–17%	19–65%
B-cells	10–20%	6–29%	1–2%	4–11%
Neutrophils	1–3%	<1–3%	30–50%	<1–2%
NK cells	1.1–1.2%	<1–2%	1–1.3%	<1–5%

*% of CD45+ cells.

## Data Availability

No new data were generated.

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
