# Peer review of "Understanding Atherosclerotic Plaque Cellular Composition: Recent Advances Driven by Single Cell Omics"

_cells, 2025, doi:10.3390/cells14110770_

Round 1
Reviewer 1 Report
Comments and Suggestions for Authors
The review article provides a very good and comprehensive overview of cell types discovered in atherosclerotic plaques by means of single cell omics studies. What is of particular interest is that it shows differences and discrepancies as well as similarities between the different published reports in the field. The individual major cell types are described in a structured manner focusing on macrophages, T-cells, B-cells and non-immune cells.
In general, it is certainly a valuable overview for the community in that area – but sometimes it reads a bit like a list, which is probably hard to avoid with this topic.
Suggestions for improvements:
- It might be good to elaborate a bit more on non-immune cells involved in atherosclerosis (working out a bit in more detail what is known about endothelial and smooth muscle cells and their changes in phenotype or cellular transitions in the course of atherosclerosis.
- It would be nice to find some epigenetic aspects of cellular identities described in the manuscript.
- A meaningful extension of table 1 might be a side-by-side comparison of human and mouse studies (describing the various cells of plaques). In this context, it would be important to emphasize that leukocytes in the circulation differ significantly between mice and humans (with significantly higher percentages of lymphocytes in mice, and higher percentages of neutrophils in humans).
- It might be advisable to elaborate a bit on perivascular fat and potential crosstalk with cells of the vasculature (e.g., SMCs) as well as infiltrating leukocytes.
- Given the difficulty of cell isolation using digestion of arteries (and potential changes of cells during the isolation procedure), it would be advisable to include articles on spatial transcriptomics (e.g., considering the following articles: https://doi.org/10.1016/j.jacc.2023.04.008; https://doi.org/10.3389/fcvm.2025.1510082; https://doi.org/10.1161/ATVBAHA.123.320330; https://doi.org/10.1038/s44161-024-00582-1; https://doi.org/10.3390/ijms26051949, and linking them to newer articles of scRNA-seq such as: https://doi.org/10.1016/j.celrep.2023.113380)
Author Response
The reviewer's original comments appear below in normal font and the authors response is indicated by bold font. Modifications for this reviewer appear in purple in the newly submitted version.
Suggestions for improvements:
- It might be good to elaborate a bit more on non-immune cells involved in atherosclerosis (working out a bit in more detail what is known about endothelial and smooth muscle cells and their changes in phenotype or cellular transitions in the course of atherosclerosis. We have reorganised the non-immune cell section to include sub-sections focused on ECs, SMCs, and regional control of cellular states. The SMC section has been expanded (line 592-606), including notably a discussion around the reference proposed by the reviewer below (https://doi.org/10.1016/j.celrep.2023.113380)
- It would be nice to find some epigenetic aspects of cellular identities described in the manuscript. Providing an exhaustive review about the epigenetics aspects of atherosclerosis is clearly beyond the scope of this review, and we feel that ad hoc mentions of some aspects of epigenetics would not be appropriate. However, we agree with the reviewer that a better description of epigenetic processes, at the single cell level, would likely provide a leap in our understanding of the pathophysiology of atherosclerosis. A paragraph to this effect has been added to the Discussion (line 804-810).
- A meaningful extension of table 1 might be a side-by-side comparison of human and mouse studies (describing the various cells of plaques). In this context, it would be important to emphasize that leukocytes in the circulation differ significantly between mice and humans (with significantly higher percentages of lymphocytes in mice, and higher percentages of neutrophils in humans). See Table 2 and addition to the text (line 184-190)
- Given the difficulty of cell isolation using digestion of arteries (and potential changes of cells during the isolation procedure), it would be advisable to include articles on spatial transcriptomics (e.g., considering the following articles):
The below articles are now cited in the review
https://doi.org/10.1016/j.jacc.2023.04.008; https://doi.org/10.1038/s44161-024-00582-1: Discussion section, line 786-797
https://doi.org/10.1016/j.celrep.2023.113380: Non-immune cell section, line 581-583 and 592-606
Reviewer 2 Report
Comments and Suggestions for Authors
This is a comprehensive review of the components of the atherosclerotic plaque mainly in humans or mice (models) using flow cytometry and scRNA analysis. What is the virtue of these modes of analysis? Cellular components so characterized are seen as clusters. How do these clusters interact with each other? Do cells within each cluster auto-react? Do cells clustered interact with cells in other clusters? Since these analyses are mainly conducted in dissociated cells of the aorta or carotid artery, they do not place the cell components within the plaque. To do this immunocytochemistry or laser captured cells are required. These aspects need to be placed in context for the purposes of this review.
While the descriptions are extensive, the operative mechanisms are only suggestive and not compelling.
Although humans and mice are assessed, the observations are not strictly comparable especially with stage of development of the process of atherosclerosis or regression.. The observation in murine lesions are not translatable to the human situation. With respect to murine processes there are limitations related to the duration of experimental diets and the ages of the sampled mice. These are important confounding factors. In particular, ATLO is only seen in aged apo E deficient mice and then largely in relation to lesions of the abdominal aorta. The basis for this selective location is only briefly addressed. The cellular components of the ATLO is addressed.
Comments on the Quality of English LanguageIn what follows suggested inserts are capitalized, while suggested deletions are in parenthesis. Sentences in many cases could be divided to shorter ones.
Line 119. "profile (more recently. of--)SIMILAR to THAT described---"
line 128 " chow diet FED MICE"
line 176 "(m0st) preferentially--"
Too many sentences begin "interestingly"
Line 258 Rephrase for clarity.
Lines 279-283 Rephrase for clarity.
Lines 296-300 Rephrase for clarity.
Line 315 "was greatly reduced in healthy aortas" reduced compared to what?
Line 318 HFD in which model?
Line 361 "not associated WITH (to) a specific ---"
Lines 365-370 rewrite for clarification
Lines 395 and 397 " T cells from symptomatic subjects. " rather than shorthand symptomatic T cells
Line 403 "initiateD"
Lines 558-9 . Clarify
Lines 565-568 rewrite for clarity. "in turn(s)"
Line 574 "SMC markers" not SCM
Line 586 " a major role OF--"
Line 637 "(un)Inconceivable"
Line 662 " were analyZED(sis)"
Author Response
The reviewer's original comments appear below in normal font and the authors response is indicated by bold font. Modifications for this reviewer appear in dark orange in the newly submitted version of the manuscript.
- What is the virtue of these modes of analysis? A small paragraph was added to the introduction to highlight the advantages of the reviewed techniques (lines 47-50)
- Cellular components so characterized are seen as clusters. How do these clusters interact with each other? Do cells within each cluster auto-react? Do cells clustered interact with cells in other clusters? Findings from cell: cell interaction technologies are discussed the manuscript, notably at lines 480-490 and 670-675. However, as stated in the discussion (line 689), an exhaustive description of these findings cannot be given here, due to the space needed to properly cover this issue. Cell: cell communication in atherosclerosis should be the subject of a separate review.
- Since these analyses are mainly conducted in dissociated cells of the aorta or carotid artery, they do not place the cell components within the plaque. To do this immunocytochemistry or laser captured cells are required. These aspects need to be placed in context for the purposes of this review. The importance of location within the plaque is discussed at several points in the manuscript, notably the preferential distribution of B-cells to adventitial ATLOs (line 128, line 488-490) and the differential transcriptomic profiles of SMC and ECs at the core of the plaque versus the proximal adjacent region (line 607-661). In response to the reviewer’s comment, we have now created a new subsection that highlights the importance of location on EC/SMC transcriptomic profiles functions (line 607, 3.4.3. Regional segregation of transcriptomic profiles). These findings were made using micro-dissected tissues, as mentioned by the reviewer.
In addition, the need to obtain information about location as part of single-omic strategies is discussed in the Discussion section, and has been expanded (line 786-797) and in situ spatial transcriptomics technologies mentioned.
- Although humans and mice are assessed, the observations are not strictly comparable especially with stage of development of the process of atherosclerosis or regression.. The observation in murine lesions are not translatable to the human situation. With respect to murine processes there are limitations related to the duration of experimental diets and the ages of the sampled mice. These are important confounding factors. In particular, ATLO is only seen in aged apo E deficient mice and then largely in relation to lesions of the abdominal aorta. The basis for this selective location is only briefly addressed. The cellular components of the ATLO is addressed. While we agree with reviewer that there are limitations to the use of animal models of atherosclerosis, we fell that this discussion is beyond the scope of this review. These limitations have been extensively discussed elsewhere, and a paragraph to that effect has been added (Lines 94-98). In addition, findings made in mice are compared to findings made in humans throughout the review, and we already highlight the use of a different species as a potential cause for apparent discrepancies (discussion, line 715). Despite the limitations of in vivo models, several key findings made in mice were validated in humans, as discussed in the review (Table 2, Figure 1, lines 184-190, lines 292-293), confirming that findings made in animal models are often useful, especially when care is taken to validate them using human samples.
- Comments on the Quality of English Language
All suggested changes have been made, including:
Shortening of sentences: Sentences were shortened at lines 180, 295, 330,681
Too many sentences begin "interestingly": Occurrences were removed from lines 114, 122, 133, 216, 222, 388, 641
line 176 "(m0st) preferentially--": Several of the cell types described at that point in the text were preferentially increased in plaque compared to blood. Thus, the suggested change “CD8+ T-cells were preferentially increased in plaque compared to blood” could be confusing. However, to simplify the sentence, “preferentially” was removed, reading “CD8+ T-cells were the most increased in plaque compared to blood” (line 180)
Line 318 HFD in which model? In ApoE-/- mice, which are the subject of the particular paragraph and of the study being discussed. Given that it is now clearly stated in line 330, we have not specified again in line 332 (former line 318) to avoid repetitions.
Text clarifications: lines 272, 295-297, 310-314, 379-382, 571-572
Reviewer 3 Report
Comments and Suggestions for Authors This is an excellent review, and I greatly appreciate the insights it provides. The composition of atherosclerotic plaques has been a subject of ongoing debate, and single-cell sequencing has brought new evidence to this field. The authors have done a commendable job summarizing the advancements in this area, and the figures are also visually appealing. A few minor suggestions: 1. The review highlights how single-cell sequencing has enhanced our understanding of plaque composition. I recommend that the authors dedicate a specific section in the discussion to explore how these new insights contribute to a better understanding of atherosclerosis and its treatment. 2. Although I recommend accepting this paper, I suggest that the authors carefully check for grammatical issues and typographical errors, such as line 733. Additionally, the abbreviation list should be arranged in alphabetical order.Author Response
The reviewer's original comments appear below in normal font and the authors response is indicated by bold font. Modifications for this reviewer appear in light blue in the newly submitted version of the manuscript.
- The review highlights how single-cell sequencing has enhanced our understanding of plaque composition. I recommend that the authors dedicate a specific section in the discussion to explore how these new insights contribute to a better understanding of atherosclerosis and its treatment. Added at lines 691-707
- Although I recommend accepting this paper, I suggest that the authors carefully check for grammatical issues and typographical errors, such as line 733. This has been addressed
- Additionally, the abbreviation list should be arranged in alphabetical order. This has been addressed